# Ingenol-3-Angelate Enhances the B Cell Inhibitory Potential of Mesenchymal Stem Cells, Leading to Marked Alleviation of Lupus Symptoms in MRL.*fas*^lpr^ Mice

**DOI:** 10.3390/ijms252312625

**Published:** 2024-11-25

**Authors:** Hong Kyung Lee, Hwa Kyung Kim, Ji Yeon Kim, Ji Su Kim, JinKyung Park, Min Sung Kim, Tae Yong Lee, Key-Hwan Lim, Hanseul Park, Dong Ju Son, Jin Tae Hong, Sang-Bae Han

**Affiliations:** 1College of Pharmacy, Chungbuk National University, Chungbuk 28160, Republic of Korea; hklee@csco.co.kr (H.K.L.); cinsia128@gmail.com (H.K.K.); kjiy8675@naver.com (J.Y.K.); woojoo_818@naver.com (J.S.K.); jkpark@csco.co.kr (J.P.); mskim1013@csco.co.kr (M.S.K.); tylee@csco.co.kr (T.Y.L.); khlim@chungbuk.ac.kr (K.-H.L.); hanpark@chungbuk.ac.kr (H.P.); sondj1@chungbuk.ac.kr (D.J.S.); jinthong@chungbuk.ac.kr (J.T.H.); 2Bioengineering Institute, CorestemChemon Inc., Gyeonggi 13486, Republic of Korea

**Keywords:** systemic lupus erythematosus, mesenchymal stem cells, B cells, ingenol-3-angelate

## Abstract

Systemic lupus erythematosus (SLE) is a chronic autoimmune disease characterized by autoantibody production by hyper-activated B cells. Although mesenchymal stem cells (MSCs) relieve lupus symptoms by inhibiting mainly T cells, whether MSCs also inhibit B cells has been controversial. Here, we found that naïve MSCs inhibited IFN-γ production by T cells, but not IgM production by B cells. We used a chemical approach to prime MSCs to inhibit B cells. We found that ingenol-3-angelate (I3A), a non-tumor-promoting phorbol ester, activated MSCs to inhibit B cells in a TGF-β1-dependent manner. We also showed that IL-1β induced MSCs to continuously secrete TGF-β1, which directly inhibited IgM production by B cells, whereas IL-1β did not. I3A-treated MSCs were better than naïve MSCs at ameliorating SLE symptoms in MRL.*fas*^lpr^ mice. In summary, our data provide information on how to generate MSCs that are effective for the treatment of SLE characterized by excessive B cell activation.

## 1. Introduction

Systemic lupus erythematosus (SLE) is an autoimmune disease that causes irreversible damage to multiple organs, such as the kidneys, skin, joints, and central nervous system [1]. Although the abnormal activation of diverse subsets of immune cells is involved in the progression of SLE, the hyper-activation of B cells might play a crucial role [2]. After the loss of self-tolerance, autoreactive T cells stimulate B cells to produce autoantibodies. These antibodies form immune complexes that are deposited throughout the body, leading to inflammation and organ damage [3]. The current treatments rely on anti-inflammatory medications, such as the anti-malarial hydroxychloroquine, glucocorticoids (including prednisone), immunosuppressants (including mycophenolate mofetil), and other biologics (including belimumab), which reduce the number of circulating B cells [4]. However, these drugs may exhibit significant side effects, such as gastrointestinal bleeding from ulcers and myocardial infarction, with a high number of patients being refractory [2,4].

Mesenchymal stem cells (MSCs) are multipotent adult stem cells and have been explored as an alternative for the treatment of SLE [4]. As to the underlying mechanisms, MSCs mainly inhibit the proliferation of and cytokine production by T cells by secreting immunosuppressive soluble factors, such as TGF-β1, indoleamine 2,3-dioxygenase 1 (IDO1), PGE2, IL-10, and nitric oxide [5,6]. In addition, PD-L1 and FasL on the surface of MSCs induce the apoptosis of T cells [7,8,9]. MSCs also inhibit the functions of dendritic cells, natural killer cells, and Th17 cells and increase the differentiation of Tregs and M2 macrophages [1]. However, whether MSCs inhibit B cells has been controversial. Some studies have shown that MSCs inhibit the proliferation of and antibody production by B cells, but others did not find such effects [10,11,12,13,14,15]. Given the crucial role of B cells in SLE progression, developing MSCs that strongly inhibit B cells would be highly beneficial. 

We previously showed that, although naïve MSCs could not inhibit B cell functions, phorbol 12-myristate 13-acetate–activated MSCs (PMA-MSCs) did inhibit B cells in CXCL-12- and PD-L1-dependent manners [16]. In addition, PMA-MSCs demonstrated superior therapeutic effects compared to naïve MSCs in MRL.*fas*^lpr^ mice [16]. However, using PMA to prime human MSCs in clinical applications may be challenging due to its tumor-promoting properties [17]. In this study, we suggest another candidate chemical to enhance the B cell inhibitory potential of MSCs: we found that ingenol 3-angelate (I3A), a non-tumor-promoting phorbol ester, enhanced the B cell inhibitory activity toward MSCs and their therapeutic effects in MRL.*fas*^lpr^ mice. We suggest here that priming of MSCs with I3A might be useful to generate clinically useful MSCs for the treatment of SLE patients.

## 2. Results

### 2.1. Effect of I3A on MSCs

First, we examined the effect of naïve MSCs on T and B cells. Naïve MSCs inhibited IFN-γ production by anti-CD3/anti-CD28 antibody-treated T cells (Figure 1A), but did not affect IgM production by B cells treated with CpG-oligodeoxynucleotide (ODN) (Figure 1B). When MSCs were pre-treated with I3A for 24 h (called I3A-MSCs), they inhibited IgM production by ODN-treated B cells (Figure 1C). Next, we investigated whether I3A-MSCs inhibited IgM production in a soluble factor- or cell–cell contact-dependent manner. I3A-MSCs suppressed IgM production by B cells regardless of whether we allowed their contact with B cells or separated the two cell types by using transwell plates (Figure 1D), suggesting that MSCs inhibited B cells in a soluble factor-dependent manner. Then, we attempted to identify the immunosuppressive factors produced by MSCs. separated the two cell types by using transwell plates (Figure 1D), suggesting that MSCs inhibited B cells in a soluble factor-dependent manner. 

Then, we attempted to identify the immunosuppressive factors produced by MSCs. I3A enhanced the expression of the genes for COX-2, IL-1β, and TGF-β1 in MSCs, but not that of the genes for iNOS, IL-6, IL-10, IDO1, FasL, or PD-L1, as assayed by RT-qPCR (Figure 1E). I3A also increased the production of PGE2, IL-1β, and TGF-β1, as assayed by ELISA (Figure 1F). Then, we transfected MSCs with siRNAs against COX-2, IL-1β, or TGF-β1 (Figure 1G). Although I3A-MSCs transfected with COX-2 siRNA inhibited B cells, I3A-MSCs transfected with IL-1β or TGF-β1 siRNA did not (Figure 1H), suggesting that the inhibition of B cells by I3A-MSCs is mediated by IL-1β and TGF-β1. I3A did not affect the phenotype (Figure 2A), viability (Figure 2B), or proliferation (Figure 2C) of MSCs.

### 2.2. Signaling in I3A-MSCs

Next, we examined signaling pathways in I3A-treated MSCs. The amounts of PKC-α and PKC-δ in the cytosol of MSCs decreased from 1 h and they almost disappeared 24 h after the start of I3A treatment (Figure 3A), suggesting the translocation of PKCs from the cytosol to the plasma membrane. I3A increased the nuclear level of NF-κB p65 (Figure 3B). I3A also increased the phosphorylation of ERK, but not that of p38 or JNK, in the cytosol of MSCs (Figure 3B). The PKC inhibitor Go6983, ERK inhibitor FR180204, and NF-κB inhibitor BAY11–7082 decreased the ability of I3A-MCSs to inhibit IgM production by B cells, but the STAT1 inhibitor fludarabine or STAT3 inhibitor AG490 did not (Figure 3C,D). We also showed that FR180204 and BAY11–7082 decreased the expression of the genes for IL-1β and TGF-β (Figure 3E), which were key soluble factors produced by I3A- MSCs (Figure 1H). These data suggested that I3A enhanced the production of IL-1β and TGF-β1 via PKC and ERK signaling pathways.

### 2.3. Effect of I3A-MSCs on Lupus Symptoms in MRL.Fas^lpr^ Mice

Next, we examined the therapeutic effect of I3A-MSCs in lupus-prone MRL.*fas*^lpr^ mice. Our preliminary experiments showed that MSCs at 4 × 10^6^ cells/mouse strongly reduced lupus symptoms in these mice [16]. To compare the efficacy of naïve MSCs and I3A-MSCs in this respect, we reduced their numbers to 4 × 10^4^ cells/mouse. Compared to PBS- and naïve MSC-injected mice, I3A-MSC-injected mice showed lower levels of anti-dsDNA (Figure 4A) and total IgG (Figure 4B) antibodies in serum and lower protein level in urine (Figure 4C); higher percentage of Foxp3-expressing Treg cells and lower percentage of IgG-producing CD138^+^ plasma cells (Figure 4D); and lower mRNA expression of inflammatory cytokines (IL-1β, TNF-α, IFN-α, and IFN-γ) in the spleen (Figure 4E). The average spleen weight of 22-week-old MRL.*fas*^lpr^ mice was average 570 mg (indicative splenomegaly), which was slightly reduced by naïve MSCs and significantly reduced by I3A-MSCs (Figure 4F).

In addition, I3A-MSC-injected mice showed a lower infiltration of CD3^+^ T cells and CD19^+^ B cells, lower deposition of complement C3 and IgG, and higher infiltration of Tregs in the kidney (Figure 5A). Since we injected MSCs generated from human bone marrow cells into the xenogeneic MRL.*fas*^lpr^ mice, we examined whether human MSCs inhibited mouse B cells or not. I3A- MSCs but not naïve MSCs inhibited IgM production by mouse B cells (Figure 5B). IFN-γ production by mouse T cells was inhibited by naïve MSCs and much more strongly by I3A-MSCs (Figure 5C). Overall, these data demonstrated that human I3A-MSCs ameliorated lupus symptoms by inhibiting T and B cells in MRL.*fas*^lpr^ mice. 

### 2.4. Role of IL-1β and TGF-β1

Next, we examined the roles of IL-1β and TGF-β1 secreted from MSCs in B cell inhibition. Blocking antibodies against IL-1β or TGF-β1 prevented MSCs from inhibiting B cells (Figure 6A). Recombinant TGF-β1 suppressed IgM production by B cells (Figure 6B), but recombinant IL-1β did not (Figure 6C). Anti-IL-1β blocking antibody decreased TGF-β1 production by I3A-MSCs (Figure 6D). Overall, these data suggest that I3A-MSCs produce two key molecules: TGF-β1, which directly inhibits IgM production by B cells, and IL-1β, which might enhance TGF-β1 production by I3A-MSCs. 

We treated MSCs with I3A for 24 h, thoroughly washed I3A out, and then immediately co-cultured I3A-MSCs with B cells. In this protocol, it was unclear for how long I3A-MSCs would retain their inhibitory activity on B cells. To address this question, we treated MSCs with I3A for 24 h, thoroughly washed I3A out, cultured I3A-MSCs for 24 h, 48 h, or 72 h without I3A, and then co-cultured them with B cells. Our data showed that I3A-MSCs retained their B cell inhibitory activity for up to 48 h after washing I3A but completely lost it by 72 h (Figure 6E). IL-1β prolonged the inhibitory effect of I3A-MSCs on B cells for at least up to 72 h (Figure 6F). TGF-β1 production by I3A-MSCs declined considerably by 72 h, but this decline was prevented by IL-1β (Figure 6G). However, IL-1β alone was not sufficient for MSCs to inhibit B cells (Figure 6H). Overall, these data suggest that IL-1β helps maintain TGF-β1 production by I3A-MSCs. 

### 2.5. Signaling in MSCs Treated with I3A and IL-1β

To examine the signaling in MSCs, we treated them with I3A or I3A/IL-1β for 24 h, then washed out the I3A and further cultured them for 72 h. Proteins were isolated after 24 h and 96 h of culture, and the levels of NF-κB p65 in the nuclei and PKC-α, p-ERK, ERK, p-p38, p38, p-JNK, and JNK in the cytosol were analyzed by western blotting. The amount of phospho-ERK increased at 24 h (see Figure 3) and was sustained for up to 96 h in I3A-MSCs; it was further increased at 96 h in I3A/IL-1β-MSCs, suggesting a boosting effect of IL-1β on ERK signaling (Figure 7A). PKCα was undetectable at 24 h and but became detectable at 96 h (presumably owing to translocation from the membrane to the cytosol) in both I3A- or I3A/IL-1β-MSCs, suggesting no effects of IL-1β on PKC signaling (Figure 7A). Compared to I3A-MSCs, the amount of NF-κB p65 in I3A/IL-1β-MSCs increased at 96 h, suggesting a boosting effect of IL-1β on NF-κB signaling (Figure 7A). Inhibitors of ERK and NF-κB weakened the inhibitory effects of I3A/IL-1β-MSCs on B cells (Figure 7B). Overall, these data suggest that IL-1β enhances ERK and NF-κB signaling in I3A-MSCs. 

In lupus-prone mice, I3A/IL-1β-MSCs decreased the serum levels of anti-dsDNA (Figure 7C) and total IgG (Figure 7D) antibodies more than I3A-MSCs did. These data suggest that IL-1β helps I3A-MSCs to efficiently inhibit B cells and ameliorate lupus symptoms in mice.

## 3. Discussion

Several priming methods to improve the immunomodulatory potential of MSCs have been used: conditioning in hypoxia [18], genetic modification to change the expression of IDO1, IL-10, TSG-6, TGF-β1, miRNAs, CD73, HGF, and PD-L1 [19,20,21,22,23,24], priming with inflammatory cytokines, such as IFN-γ, TNF-α, and IL-1β [25,26,27], and chemical priming with lipopolysaccharide, PMA, atorvastatin, rapamycin, and retinoic acid [16,28,29,30,31,32]. These approaches have generally increased the production of immunosuppressive molecules, including IDO1, PGE2, and TGF-β1, which inhibited the functions of T cells, dendritic cells, and M1 macrophages and showed therapeutic effects in animal models of diverse autoimmune diseases [16,28,29,30,31,32]. We have been continuously seeking chemicals to prime MSCs to enhance their inhibitory effects on B cells, which play a key role in the pathogenesis of SLE. We previously reported that IFN-γ and PMA increased the B cell inhibitory potential of MSCs and that activated MSCs showed stronger therapeutic activity than naïve MSCs in MRL.*fas*^lpr^ mice [16,32]. PMA might be stronger than IFN-γ in its ability to prime MSCs. However, PMA cannot be used to prime MSCs in clinical applications because of its tumor-promoting properties [17]. To surmount this hurdle, we screened PMA analogues and found that I3A has similar activity to that of PMA. Since I3A, a non-tumor-promoting phorbol ester, has been used for the treatment of actinic keratosis and multiple skin disorders [33,34,35], it might be safer than PMA. In this study, we found that, similar to PMA, I3A enhanced the B cell inhibitory potential of MSCs and improved their therapeutic efficacy in MRL.*fas*^lpr^ mice. I3A-MSCs inhibited B cells in a TGF-β1-dependent manner and more effectively alleviated lupus symptoms in MRL.*fas*^lpr^ mice compared to naïve MSCs. 

Although both I3A and PMA are phorbol ester analogues, the action mechanisms of I3A-MSCs and PMA-MSCs are different from each other. When separated from B cells in transwell assay, PMA-MSCs weakly inhibit B cell functions, suggesting contact- and soluble factor-dependent inhibition [16]. PMA-MSCs attract B cells by secreting CXCL-10 and induce B cell apoptosis in a PD-L1-dependent manner. PMA-MSCs also inhibit B cells using TGF-β1 and PGE2 [16]. However, we found in this study that I3A-MSCs strongly inhibit B cells in transwell assay, suggesting soluble factor-dependent inhibition only. I3A-MSCs inhibited B cells by using two soluble factors, TGF-β1 and IL-1β. Overall, our previous and current data suggest that the mechanisms of action of effector MSCs vary depending on the chemical activators used.

We examined the roles of TGF-β1 and IL-1β. By using siRNAs, recombinant proteins, and neutralizing antibodies, we demonstrated that TGF-β1 directly inhibits B cell function, consistent with previous reports [5]. However, IL-1β did not directly inhibit the IgM production by B cells. These data are consistent with a previous observation that the antibody production by and proliferation of B cells in response to mitogenic stimuli are normal in IL-1β-knockout mice [36]. However, IL-1β might affect other functions of MSCs. IL-1β increases their proliferation and chondrogenic potential [37]. Pre-activation of MSCs with IL-1β enhances its paracrine effects in a radiation-induced intestinal injury model via a heme oxygenase-1-dependent mechanism [38]. Conditioned medium of MSCs pre-activated with IL-1β prevents apoptosis and promotes the proliferation of epithelial cells, enhances the regeneration of intestinal stem cells, and down-regulates radiation-induced inflammatory responses at systemic and mucosal levels [38]. In this study, we documented the previously unrecognized effect of IL-1β on MSCs: IL-1β prolonged TGF-β1 production by I3A-MSCs. This observation is crucial for developing clinically effective MSCs. Typically, SLE patients would receive pure I3A-MSCs after thorough washing to remove I3A. However, predicting how long I3A-MSCs will continue producing TGF-β1 remains challenging. To address this point, we treated MSCs with I3A from for 24 h, then washed out I3A and cultured them for an additional 72 h. By 96 h, I3A-MSCs had lost their ability to produce TGF-β1, potentially reducing their *in vivo* efficacy. However, when I3A was supplemented with exogenous IL-1β, they continued to produce TGF-β1 for 96 h from the start of treatment, indicating an enhancing effect of IL-1β. Overall, our data suggest that I3A and IL-1β may synergistically enhance the B cell inhibitory potential of MSCs. 

The implications of our study are limited by several caveats. First, we did not clarify the signaling in I3A- and IL-1β-MSCs in detail. I3A activates a broad range of PKC isoforms, NF-κB, ERK, p38, and AP-1, which are also activated by IL-1β [38,39]. Although we demonstrated the activation of PKC, NF-κB, and ERK signaling, it will be interesting to examine the signaling cross-talk between I3A and IL-1β in MSCs. Second, we identified TGF-β1 as a key soluble factor among seven soluble factors secreted from I3A-MSCs. However, MSCs secrete multiple soluble factors, such as cytokines, chemokines, growth factors, extracellular matrix components, and metabolic products [40]. Thus, it will be interesting to identify other B cell inhibitory soluble factors derived from I3A-MSCs. Third, I3A has cell-type specific effects. It inhibits the growth of cancer cells at micromolar concentrations and induces apoptosis in breast, lung, leukemic, melanoma, colon and prostate cancer cells [41,42,43]. On the other hand, I3A enhances the functions of normal cells at nanomolar concentrations. It activates endothelial cells to express E-selectin, ICAM-1, and IL-8, triggering the recruitment of tumoricidal neutrophils [44]. I3A also enhances the functions of cytotoxic T cells and natural killer cells [41,45]. It will be interesting to compare the effects of I3A on MSCs, immune cells, and cancer cells at different concentrations. Fourth, it will be interesting to examine the effect of exosomes isolated from I3A-MSCs, since exosomes and their parental cells have similar biological effects [46]. It is plausible that exosomes derived from I3A-MSCs might include more TGF-β1 than those from naïve MSCs.

Despite these shortcomings, our data demonstrate that I3A might be a good tool compound to prime MSCs to strongly inhibit B cells and to ameliorate the progression of SLE.

## 4. Materials and Methods

### 4.1. Generation of MSCs

MSCs were generated from bone marrow cells aspirated from the posterior iliac crest of healthy donors. Mononuclear cells were purified by density gradient centrifugation (Ficoll-Paque; GE Healthcare Bio-Sciences AB, Uppsala, Sweden) and cultured at 2 × 10^7^ cells/T175 flask in CSBM-A06 medium (Corestem Inc., Gyeonggi, Korea) with 10% fetal bovine serum (BD Biosciences, Franklin Lakes, NJ, USA), 2.5 mM L-glutamine, and penicillin/streptomycin (Sigma-Aldrich, St. Louis, MO, USA) in an incubator with 5% CO_2_ maintained at 37 °C. Medium was changed every 3 or 4 days and MSCs were used in experiments at passage 3–5. MSCs did not express markers such as CD34 or HLA-DR, and were positive for CD73, CD90, and CD105. MSCs were treated with I3A (1 μg/mL) for 24 h, washed three times with medium, and used in all experiments [7]. MSCs (4 × 10^4^ cells/mouse, *n* = 5) were intraperitoneally injected into the mice at the age of 12 weeks.

### 4.2. Isolation of B and T Cells

Human peripheral blood mononuclear cells were obtained from Zen-Bio (Research Triangle, NC, USA). They were rapidly thawed and washed with PBS. Human B and T cells were purified using a human B and T cell isolation kits, respectively (Miltenyi Biotec, Auburn, CA, USA). Human B and T cells were used all experiments, except for Figure 5B,C, in which mouse B (mB) and T (mT) cells were used. mB and mT cells were purified from the spleen of MRL.*fas*^lpr^ mice by a negative depletion method using a mouse B and T cell isolation kit (Miltenyi Biotec, Auburn, CA, USA) [32]. The purity levels are shown in Appendix A.

### 4.3. ELISA, RT-qPCR, ³[H]-Thymidine Incorporation Assay and Western Blotting

B cells and MSCs were co-cultured for 72 h in the presence of ODN (1 μg/mL), and the concentrations of IgM in culture medium were measured by ELISA (Bio-Techne, Minneapolis, MN, USA). T cells and MSCs were co-cultured for 72 h in the presence of anti-CD3 antibody (1 μg/mL) and anti-CD28 antibody (1 μg/mL) (R&D Systems, Minneapolis, MN, USA). The concentrations of IFN-γ in culture medium were measured by ELISA (Bio-Techne, Minneapolis, MN, USA). MSCs alone were cultured for 24 h and the concentrations of IL-1β, PGE2, and TGF-β1 were measured by using ELISA kits (Bio-Techne, Minneapolis, MN, USA) [16].

Total RNA was extracted from MSCs using Trizol Reagent (Thermo Fisher Scientific, Waltham, MA, USA). cDNA was synthesized from 1 μg total RNA using an RT premix kit (Bioneer, Daejon, Korea). Quantitative PCR was performed with SYBR Green Master Mix (Elpis Biotech., Daejon, Korea) in a StepOnePlus Real-Time PCR System (Applied Biosystems, Foster City, CA, USA). Relative mRNA levels in a sample were based on its threshold cycle in comparison with that of the housekeeping gene for β-actin [47]. The primer sequences are shown in Appendix A. 

To measure cell proliferation, MSCs (1 × 10^4^ cells/well) were seeded in 96-well plates. The cells were pulsed with ³H-thymidine (113 Ci/mmol; NEN, Boston, MA, USA) at a concentration of 1 μCi/well for 24 h using an automated cell harvester (Inotech, Dottikon, Switzerland). The amount of ³H-thymidine incorporated into the cells was measured using a Wallac MicroBeta scintillation counter (Wallac, Turku, Finland).

MSCs were lysed on ice for 10 min with cell lysis buffer according to the manufacturer’s instruction (Cell Signaling Technology, Danvers, MA, USA). Proteins were separated by 10% SDS-PAGE and transferred to a polyvinylidene fluoride membrane (Millipore-Sigma, Burlington, MA, USA). Next, the membrane was blocked with 5% skim milk in TBS/Tweem-20 (TTBS) for 1 h. Subsequently, primary antibodies in TTBS containing 5% BSA were incubated with the blocked membranes overnight at 4 °C. Antibodies against human PKC-α, PKC-δ, GAPDH, phospho-ERK, ERK, phospho-p38, p38, phospho-JNK, JNK, p65, and H-1 were purchased from Cell Signaling Technology (Danvers, MA, USA). After washing, membranes were incubated with horseradish peroxidase–conjugated secondary antibody (Cell Signaling Technology, Danvers, MA, USA) and detected by using a ChemiDoc XRS+ system (BioRad, Hercules, CA, USA) [32].

### 4.4. RNA Interference

Small interfering RNAs (siRNAs) were purchased from Bioneer (Daejon, Korea). The following sequences were used: COX-2, 5’-CAC CAA GAG UAU AAA CCU U-3’, 5’-GGA CUG CUA UUU AGC UCC U-3’, 5’-CAG AUG AAA UAC CAG UCU U-3’; IL-1β, 5’-CAC GAU GCA CCU GUA CGA U-3’, 5’-UCU ACA GCU GGA GAG UGU A-3’, 5’-GUU UUU GAG UAC GGC UAU A-3’; TGF-β1, 5’-CAG AGU ACA CAC AGC AUA U-3’, 5’-CGC GUG CUA AUG GUG GAA A-3’, 5’-GAC AUC AAC GGG UUC ACU A-3’. The following negative-control siRNA was used: 5′-CCU ACG CCA CCA AUU UCG U-3′. MSCs were transfected with 100 pM siRNAs using Lipofectamine RNAiMAX reagent (Thermo Fisher Scientific, Waltham, MA, USA) following the manufacturer’s protocol and were incubated at 37 °C in a 5% CO_2_ incubator for 24 h [48].

### 4.5. Lupus-Prone MRL.Fas^lpr^ Mouse Model

MRL.MpJ-*Tnfrsf6^Faslpr^*/J (called MRL.*fas*^lpr^ hereafter) mice lack *Fas* and spontaneously develop an SLE-like disease [7]. The onset and symptom severity in these mice depend on their genetic background. Female MRL.*fas*^lpr^ mice die at an average age of 17 weeks and males die at 22 weeks. Similar to SLE patients, MRL.*fas*^lpr^ mice have a marked increase in anti-dsDNA antibodies in their blood and develop severe nephritis [7]. Female MRL.*fas*^lpr^ mice were purchased from the Jackson Laboratory (Bar Harbor, ME, USA). Mice were housed in specific pathogen-free conditions at 21–24 °C and 40–60% relative humidity under a 12 h light/dark cycle and randomized into 3 groups. Mice were injected intravenously with PBS (vehicle, *n* = 5), naïve MSCs (4 × 10^4^ cells/mouse, *n* = 5), or I3A-MSCs (4 × 10^4^ cells/mouse, *n* = 5) once at the age of 12 weeks. Serum and urine were obtained at 22 weeks. The levels of anti-dsDNA IgG and total IgG in serum and the levels of protein in urine were measured by using ELISA kits purchased from Alpha Diagnostic International (San Antonio, TX, USA), eBioscience (San Diego, CA, USA), and Sigma-Aldrich (St. Louis, MO, USA), respectively. 

### 4.6. Phenotype Analysis

Spleens were isolated from MRL.*fas*^lpr^ mice at 22 weeks of age. For phenotyping, spleen cells were stained with APC-conjugated antibody against mouse CD4 or CD138 (BD Biosciences, Franklin Lakes, NJ, USA). Cells were fixed and permeabilized using a Cytofix/Cytoperm Kit (BD Biosciences, Franklin Lakes, NJ, USA) and then stained with anti-Foxp3-PE or anti-IgG-FITC antibody (eBioscience, San Diego, CA, USA). MSC viability was examined by the propidium iodide (PI) uptake assay [49]. The cells were stained with 1 μg/mL of PI and were analyzed with a FACSCalibur flow cytometry. The cells stained with PI were considered dead. Cells were examined by flow cytometry (FACSCalibur, BD Biosciences, Franklin Lakes, NJ, USA) and the data were analyzed in CellQuest Pro software (BD Biosciences, Franklin Lakes, NJ, USA) [50]. 

### 4.7. Immunohistochemistry

Kidneys were isolated from the surviving MRL.*fas*^lpr^ mice at 22 weeks of age and fixed with 10% formalin for 24 h. After dehydration with ethanol and xylene, the tissues were embedded in paraffin and cut into 4 μm sections. After removing paraffin, sections were hydrated and heated in a microwave oven (650 W for 20 min) for antigen retrieval, after which endogenous peroxidase activity was blocked with 3% hydrogen peroxide. To detect immune cells in the kidney, sections were incubated with the primary goat antibodies against CD3 (1:100; Santa Cruz Biotechnology, Dallas, TX, USA), CD19 (1:100; BioLegend, San Diego, CA, USA), C3 complement (1:100; GeneTex, San Antonio, TX, USA), mouse IgG (diluted 1:100; Jackson ImmunoResearch, West Grove, PA, USA), and Foxp3 (1:50; Abcam, Cambridge, UK) at 4 °C overnight. The sections were then incubated with secondary antibody (anti-goat IgG conjugated with horseradish peroxidase; Vector Laboratories, Burlingame, CA, USA) for 1 h at room temperature. Signals were developed with a two-component high-sensitivity diaminobenzidine chromogenic substrate (Vector Laboratories) for 10 min and the sections were counter-stained with hematoxylin. Stained area (%) was calculated with ImageJ software (NIH, Bethesda, MD, USA) as IHC-stained area (brown) / total area (brown + non-brown) [16].

### 4.8. Statistical Analysis

Data are presented as the mean ± SEM of at least three independent *in vitro* experiments performed in triplicate or of five mice. To determine statistical significance, *p*-values were calculated using one-way ANOVA (GraphPad Software, San Diego, CA, USA) [51].

## 5. Conclusions 

The results of this study provide several insights into the therapeutic mechanisms of MSCs in SLE. First, we verified that naïve MSCs ameliorate lupus symptoms in MRL.*fas*^lpr^ mice by inhibiting T cells only. Second, we found that I3A-MSCs strongly ameliorate SLE symptoms in mice by inhibiting B cells as well as T cells. Third, I3A-MSCs inhibit B cells in a TGF-β1-dependent manner. Fourth, IL-1β prolongs the secretion of TGF-β1 by I3A-MSCs. Our data provide a clue on how to generate MSCs effective for treatment of SLE characterized by excessive B cell activation.

## Figures and Tables

**Figure 1 ijms-25-12625-f001:**
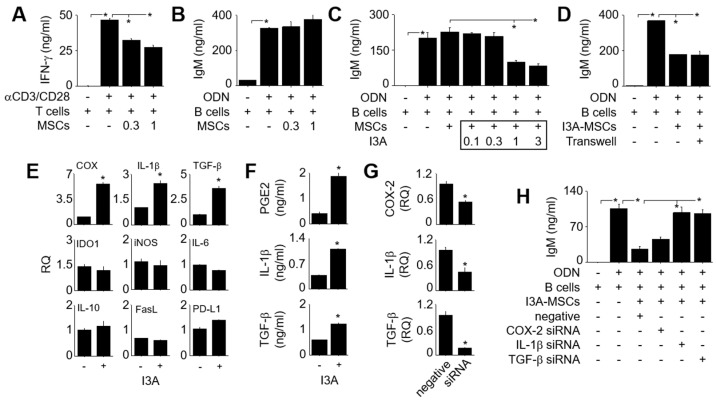
**Effect of I3A on the immunosuppressive functions of MSCs.** (**A**) MSCs (0.3–1 × 10^3^ cells/well) and T cells (100 × 10^3^ cells/well) were co-cultured for 72 h, and T cells were stimulated with anti-CD3 and anti-CD28 antibodies (1 μg/mL each). The concentration of IFN-γ in the medium was measured by ELISA. (**B**) MSCs (0.3–1 × 10^3^ cells/well) and B cells (100 × 10^3^ cells/well) were co-cultured for 72 h, and B cells were stimulated with CpG-oligodeoxynucleotide (ODN, 5 μg/mL). The concentration of IgM in the medium was measured by ELISA. (**C**) MSCs were pre-treated with I3A (0.1 − 3 μg/mL) for 24 h and washed with medium (I3A–MSCs). I3A–MSCs (1 × 10^3^ cells/well) and B cells (100 × 10^3^ cells/well) were co-cultured for 72 h, and the concentration of IgM in the medium was measured by ELISA. (**D**) I3A-MSCs (5 × 10^3^ cells/well) were added to the lower wells and B cells (500 × 10^3^ cells/well) to the upper wells of a transwell (pore size: 0.4 μm). The concentration of IgM in the medium was measured by ELISA. (**E**,**F**) Total RNA was isolated from control MSCs ( − ) or I3A-MSCs ( + ). Gene expression levels were measured by RT-qPCR (**E**). The concentrations of IL-1β, PGE2, and TGF–β1 accumulated in the medium over 24 h were assessed by ELISA (**F**). (**G**) I3A–MSCs were transfected with negative–control, COX–2, TGF-β1, or IL–1β siRNA for 24 h, and the levels of the corresponding transcripts were measured by RT–qPCR. (**H**) I3A-MSCs (1 × 10^3^ cells/well) transfected with siRNAs as in (**G**) were co-cultured with B cells (100 × 10^3^ cells/well) for 72 h. The concentration of IgM in the medium was measured by ELISA. * *p* < 0.01 (*n* = 3).

**Figure 2 ijms-25-12625-f002:**
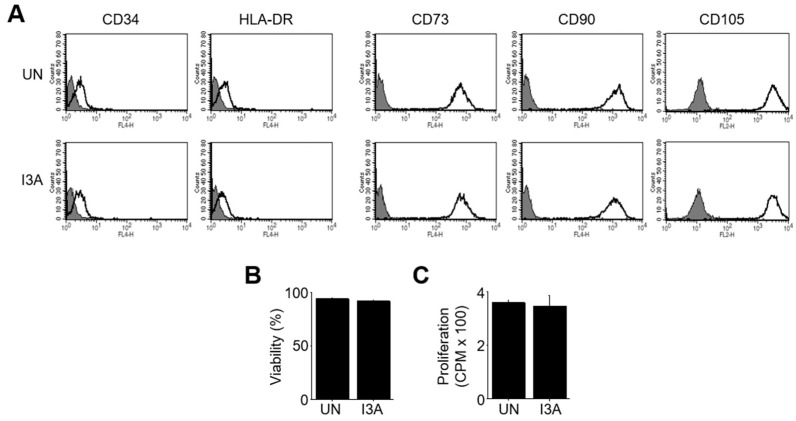
**Effect of I3A on the phenotype, viability, and proliferation of MSCs.** MSCs were treated with I3A (3 μg/mL) for 24 h and washed with medium. (**A**) Phenotype of MSCs was examined by flow cytometry. (**B**) Viability of MSCs was examined with propidium iodide uptake assay. (**C**) Proliferation of MSCs was examined with ³[H]-thymidine incorporation assay.

**Figure 3 ijms-25-12625-f003:**
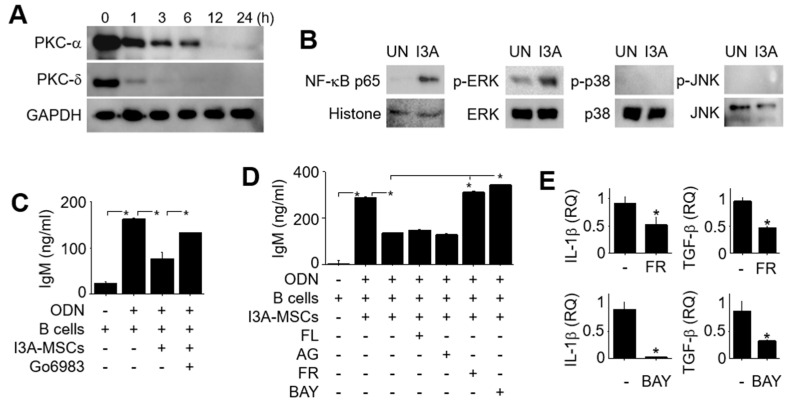
**Signaling in I3A-MSCs.** (**A**) MSCs were treated with I3A (1 μg/mL) for 24 h, and protein levels of PKC-α and PKC-δ in the cytosol were examined by western blotting. (**B**) The level of NF-κB p65 in the nuclei and the levels of p-ERK, ERK, p-p38, p38, p-JNK, and JNK in the cytosol were examined by western blotting. (**C**) MSCs were treated with I3A in the presence or absence of the PKC inhibitor Go6983 (1 μg/mL) for 24 h and then washed three times with medium. I3A-MSCs (1 × 10^3^ cells/well) were co-cultured with B cells (100 × 10^3^ cells/well) for 72 h, and the concentration of IgM in the medium was measured by ELISA. (**D**) MSCs were pre-treated with I3A (1 μg/mL) for 24 h in the presence of the STAT1 inhibitor fludarabine (FL; 100 μM), STAT3 inhibitor AG490 (AG; 100 μM), ERK inhibitor FR180204 (FR; 100 μM), or NF-κB inhibitor BAY-7082 (BAY; 1 μM) for 24 h, and then washed three times with medium. I3A-MSCs (1 × 10^3^ cells/well) were co-cultured with B cells (100 × 10^3^ cells/well) for 72 h, and the concentration of IgM in the medium was measured by ELISA. (**E**) MSCs were pre-treated with I3A (1 μg/mL) for 24 h in the presence of FR (100 μM) or BAY (1 μM) for 24 h. Expression levels of the IL-1β and TGF-β1 genes were measured by RT-qPCR. * *p* < 0.01 (*n* = 3).

**Figure 4 ijms-25-12625-f004:**
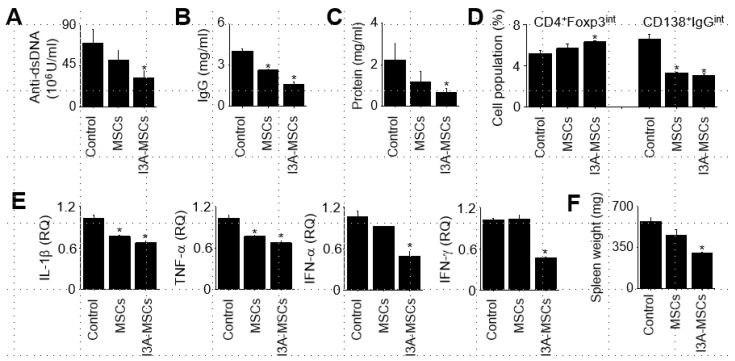
**Effect of I3A-MSCs on lupus symptoms in MRL.*fas*^lpr^ mice.** MRL.*fas*^lpr^ mice were intravenously injected with PBS (control), naïve MSCs (4 × 10^4^ cells/mouse), or I3A-MSCs (4 × 10^4^ cells /mouse) once at the age of 12 weeks. All mice were sacrificed at the age of 22 weeks. (**A**–**C**) The concentrations of anti-dsDNA IgG (**A**) and total IgG (**B**) in serum and proteinuria (**C**) were measured by ELISA at the age of 22 weeks. (**D**) The percentages of Foxp3-exrpessing Tregs (CD4^+^Foxp3^int^) and plasma cells (CD138^+^IgG^int^) in the spleens were analyzed by flow cytometry. (**E**) Total RNA was isolated from spleen cells and the mRNA levels of IL-1β, TNF-α, IFN- α, and IFN-γ were measured by RT-qPCR. (**F**) The spleen weights are shown. ^*^*p* < 0.01 (*n* = 5).

**Figure 5 ijms-25-12625-f005:**
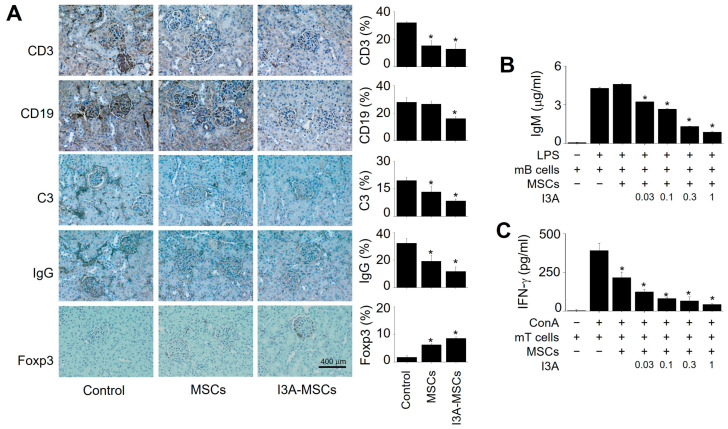
**Immunohistochemical analysis.** (**A**) Kidneys were isolated MRL.*fas*^lpr^ mice used in Figure 4. Kidney sections were stained with antibodies against CD3 (T cells), CD19 (B cells), C3 complement, IgG, or Foxp3 (Treg cells). Representative photos are shown (magnification, 100×; scale bars 400 μm) * *p* < 0.01 (*n* = 5). (**B**) MSCs were pre-treated with I3A (0.03–1 μg/mL) for 24 h. I3A-MSCs (1 × 10^4^ cells/well) and mouse B (mB) cells (10 × 10^4^ cells/well) isolated from MRL.*fas*^lpr^ mice were co-cultured for 72 h. The concentration of IgM in the medium was measured by ELISA. Lipopolysaccharide (LPS, 1 μg/mL) was used to activate B cells. (**C**) I3A-MSCs (1 × 10^4^ cells/well) and mouse T (mT) cells (10 × 10^4^ cells/well) isolated from MRL.*fas*^lpr^ mice were co-cultured for 72 h, and the concentration of IFN-γ in the medium was measured by ELISA. Concanavalin A (ConA, 1 μg/mL) was used to activate T cells (**C**). * *p* < 0.01 (*n* = 3).

**Figure 6 ijms-25-12625-f006:**
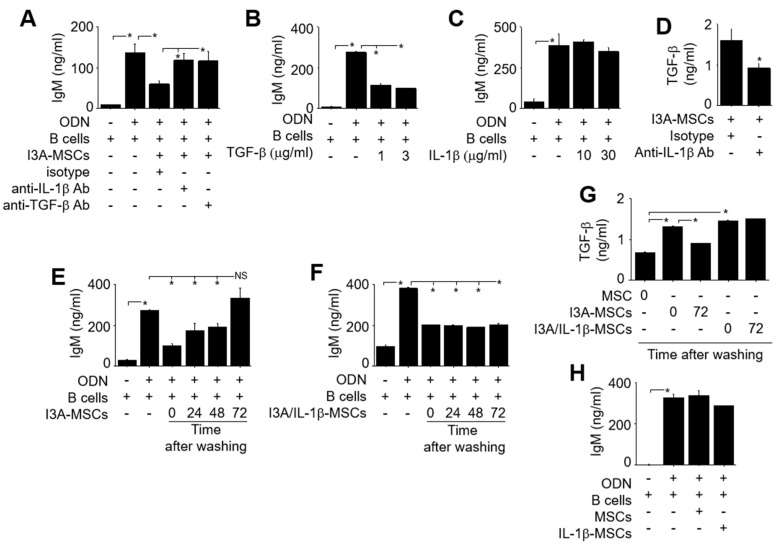
**Roles of TGF-β1 and IL-1β.** (**A**) I3A-MSCs (1 × 10^3^ cells/well) and B cells (100 × 10^3^ cells/well) were co-cultured for 72 h in the presence of blocking antibodies against IL-1β or TGF-β1. The concentration of IgM in the medium was measured by ELISA. (**B**,**C**) B cells were cultured with recombinant TGF-β1 (B; 1 or 3 μg/mL) or IL-1β (C; 10 or 30 μg/mL). The concentration of IgM in the medium was measured by ELISA. (**D**) I3A-MSCs (1 × 10^3^ cells/well) were cultured for 72 h in the presence of IL-1β blocking antibody. The concentration of TGF-β1 in the medium was measured by ELISA. (**E**) MSCs were pre-treated with I3A (1 μg/mL), washed with medium, and further incubated for 24, 48, or 72 h without I3A. Then, I3A-MSCs (1 × 10^3^ cells/well) were co-cultured with B cells (100 × 10^3^ cells/well) for 72 h. The concentration of IgM in the medium was measured by ELISA. (**F**) MSCs (named I3A/IL-1β-MSCs) were pre-treated with both I3A (1 μg/mL) and IL-1β (12.5 ng/ml), washed with medium, and further incubated for 24, 48, and 72 h. Then, I3A/IL-1β-MSCs (1 × 10^3^ cells/well) were co-cultured with B cells (100 × 10^3^ cells/well) for 72 h. The concentration of IgM in the medium was measured by ELISA. (**G**) MSCs were pre-treated with I3A alone (1 μg/mL) or with both I3A (1 μg/mL) and IL-1β (12.5 ng/ml) for 24 h, washed with medium, and further incubated for 72 h. The concentration of TGF-β1 in the medium was measured by ELISA. (**H**) MSCs were pre-treated with IL-1β (50 ng/ml) for 24 h (IL-1β-MSCs). Naïve MSCs or IL-1β-MSCs (1 × 10^3^ cells/well) and B cells (1 × 10^3^ cells/well) were co-cultured for 72 h. The concentration of IgM in the medium was measured by ELISA. * *p* < 0.01 (*n* = 3).

**Figure 7 ijms-25-12625-f007:**
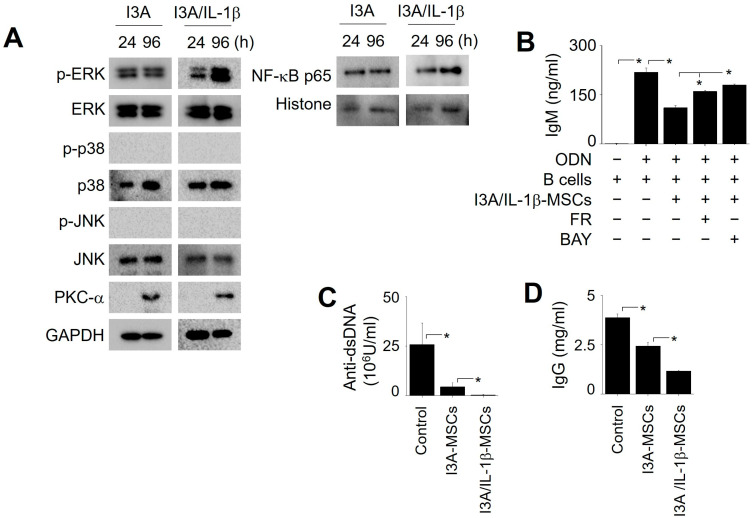
**Signaling in MSCs treated with I3A and IL-1β.** (**A**) MSCs were treated with I3A (1 μg/mL) and in the absence or presence of IL-1β (12.5 ng/mL) for 24 h, washed with medium, and further incubated for 72 h. The level of NF-κB p65 in the nuclei and those of PKC-α, p-ERK, ERK, p-p38, p38, p-JNK, and JNK in the cytosol were examined by western blotting. (**B**) MSCs were pre-treated with I3A and IL-1β in the absence or presence of the ERK inhibitor FR180204 (FR; 100 μM) or NF-κB inhibitor BAY11-7082 (BAY; 1 μM) for 24 h, and then were co-cultured with B cells for 72 h. ODN was used to activate B cells. The concentration of IgM in the medium was measured by ELISA. * *p* < 0.01 (*n* = 3). (**C**,**D**) MRL.*fas*^lpr^ mice were intravenously injected with PBS (control), I3A-MSCs (1 × 10^4^ cells/mouse), or I3A/IL-1β-MSCs (1 × 10^4^ cells /mouse) once at the age of 12 weeks. Serum was isolated at the age of 22 weeks. The levels of anti-dsDNA IgG (**C**) and total IgG antibody (**D**) were measured by ELISA. * *p* < 0.01 (*n* = 4).

## Data Availability

The datasets generated and/or analyzed during the current study are available from the corresponding author upon reasonable request.

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
