# Peer review of "Ingenol-3-Angelate Enhances the B Cell Inhibitory Potential of Mesenchymal Stem Cells, Leading to Marked Alleviation of Lupus Symptoms in MRL.faslpr Mice"

_ijms, 2024, doi:10.3390/ijms252312625_

Round 1

Reviewer 1 Report

Comments and Suggestions for Authors

The abstract is concise but effectively delivers the essential information about the study's objectives and findings. It gives readers a clear overview without overwhelming them with details, ensuring a focus on the most significant aspects of the research.

The introduction is well-structured, progressing logically from a general discussion about Systemic Lupus Erythematosus (SLE) and its pathomechanisms to a more specific focus on mesenchymal stem cells (MSCs). The authors clearly articulate the rationale behind their study and smoothly transition into stating the purpose of their research, which is essential for engaging readers and providing context.

In the methodology section, a wide range of laboratory techniques is described in detail, showing the study's scientific rigor. The comprehensive explanation of methods ensures reproducibility and gives credibility to the experimental design.

The results section is well-organized, highlighting key findings such as the effect of I3A on MSCs, with figures that visually enhance the data and make complex results easier to grasp. The subsequent section on signaling in I3A-MSCs is also written in detail, providing insights into molecular pathways, adding depth to the results. Further data regarding MSC behavior and the inclusion of laboratory experiments on mice bring additional value, illustrating the study's translational potential.

The discussion is thoughtful, offering a balanced interpretation of the results, placing them in the context of existing literature. The references are thorough and appropriately cited, supporting the arguments made throughout the manuscript. Overall, the study is well-rounded, with strong methodology, clear presentation of results, and insightful discussion. Expanding some sections, like the discussion or references, could further enhance the manuscript's depth, but it is already quite solid in its current form.

Author Response

Thank you for the reviewer’s high praise.

Reviewer 2 Report

Comments and Suggestions for Authors

The manuscript presents an interesting study on the therapeutic potential of mesenchymal stem cells (MSCs) in systemic lupus erythematosus (SLE). While the experiments are well-conducted and address relevant biological questions, there are several areas where additional clarification or data would improve the manuscript's scientific rigor and overall readability. Below are detailed comments to help refine the manuscript:

1. Line 22: Please clarify whether both IDO1 and IDO2 are being measured or if the focus is only on one of these enzymes. This clarification is crucial for understanding the scope of your study and its relevance to MSC-mediated immunomodulation.

2. Please discuss the rationale for selecting an MSC dose of 4 x 10^4 cells. Was this dose optimized based on previous studies, or is it a standard dose used in similar experimental models?

3. Section 2.5: Why was the 72-hour timepoint not included, especially given that I3A-MSCs lose their B-cell inhibitory effect by 72 hours, as per previous findings?

4. Section 4.1: Please elaborate on the timelines for MSC treatment, as some therapeutic effects are time-dependent.

5. Clarify the source of MSCs wherever applicable (e.g., human or mouse), as it appears you used both sources depending on in vitro and in vivo experiments. Explicitly mentioning the source in each relevant section will improve transparency and reproducibility.

6. Section 4.2: Please provide purity data for immune cells, such as flow cytometry profiles, as supplemental data. This is important to validate the identity and purity of the specific immune population used in your experiments.

7. Line 352: Specify the method used for lysing MSCs. This detail is critical for reproducibility, as the lysis method could influence downstream experimental outcomes.

8. Did you observe any effect of MSC treatment on splenomegaly, given that you performed phenotypic analyses of the spleens? Splenomegaly is a key feature in SLE and addressing this would add value to your study.

9. Line 222: Replace the word "better" with "more" in this context, as "more" is a more appropriate descriptor for the intended meaning.

Author Response

The manuscript presents an interesting study on the therapeutic potential of mesenchymal stem cells (MSCs) in systemic lupus erythematosus (SLE). While the experiments are well-conducted and address relevant biological questions, there are several areas where additional clarification or data would improve the manuscript's scientific rigor and overall readability. Below are detailed comments to help refine the manuscript:

Comments 1: Line 22: Please clarify whether both IDO1 and IDO2 are being measured or if the focus is only on one of these enzymes. This clarification is crucial for understanding the scope of your study and its relevance to MSC-mediated immunomodulation.

Response 1: Thank you for pointing this out. We agree with this comment. Although IDO1 and IDO2 are closely related tryptophan catabolizing enzymes, they have different immunoregulatory functions. IDO1 has anti-inflammatory roles, but IDO2 has proinflammatory roles to activate T and B cells (Frontiers in Immunology, 11, 1861, 2020). In our study, we measured the expression levels of IDO1; therefore, we have changes “IDO” to “IDO1” in the revised manuscript (line 39, 91, 243, & 247)

Comment 2: Please discuss the rationale for selecting an MSC dose of 4 x 10^4 cells. Was this dose optimized based on previous studies, or is it a standard dose used in similar experimental models?

Response 2: In our previous study, when we injected human MSCs (1 x 106 cells/injection) into the MRL.Faslpr mice at 12 weeks of age, MSCs prolonged survival: 90% of the mice receiving MSCs survived up to 28 weeks of age, whereas only 10% of the control mice survived (Scientific Reports, 7, 41258, 2017). In another previous study, we injected MSCs (1 x 104 cells/injection), prednisone (0.5 mg/kg), or both into the MRL.Faslpr mice at 12 weeks of age. The combination of MSCs and prednisone prolonged survival: 90% of mice receiving both drugs survived up to 25 weeks of age, whereas only 10% of control and MSC-treated mice and 50% of prednisone-treated mice survived (Stem cells International, 41258, 2018). In this study, to compare the efficacy of naïve MSCs and I3A-MSCs, we reduced their numbers to 4 × 104 cells/mouse. Based on our previous stdies, we optimized the doses of MSCs to 1-4 × 106 cells/mouse for therapeutic mechanism studies of MSCs and to 1-4 × 104 cells/mouse for the combination studies and MSC priming studies. According to this comment, we have highlighted this point in section 2.3 (line 132~135).

Comment 3: Section 2.5: Why was the 72-hour timepoint not included, especially given that I3A-MSCs lose their B-cell inhibitory effect by 72 hours, as per previous findings?

Response 3: Thank you for pointing this out. In our previous study, we observed that phorbol 12-myristate 13-acetate (PMA) activated MSCs to inhibit B cells; however, the B cell inhibitory activity of PMA-treated MSCs moderately diminished at 48 h, significantly at 72 h, and disappear entirely by 96 h and 120 h. In this study, the B cell inhibitory activity of I3A-treated MSCs diminished gradually but followed a similar pattern to that of PMA-treated MSCs. Therefore, we considered that 72 h might be sufficient to assess how long I3A-treated MSCs inhibit B cells.

Comment 4: Section 4.1: Please elaborate on the timelines for MSC treatment, as some therapeutic effects are time-dependent.

Response 4: In response to this comment, we have added the following sentence to the revised manuscript (line 328-329).

“MSCs (4 × 104 cells/mouse, n = 5) were intraperitoneally injected into the mice at the age of 12 weeks.”

Comment 5: Clarify the source of MSCs wherever applicable (e.g., human or mouse), as it appears you used both sources depending on in vitro and in vivo experiments. Explicitly mentioning the source in each relevant section will improve transparency and reproducibility.

Response 5: Thank you for pointing this out. We have used human MSCs as effector cells in in vivo experiments. In all in vitro experiments, we have used B and T cells isolated from human PBMCs as target cells, except in Figures 5B and 5C, where B and T cells isolated from the spleens of MRL.Faslpr mice were used. To clarify it, we have added a sentence to section 4.2. (line 334-335) as follows.

“Human B and T cells were used all experiments, except for Figure 5B and 5C, in which mouse B (mB) and T (mT) cells were used.”

We have also changed Figure 5B and 5C and its legend.

Comment 6. Section 4.2: Please provide purity data for immune cells, such as flow cytometry profiles, as supplemental data. This is important to validate the identity and purity of the specific immune population used in your experiments.

Response 6: According to this comment, we have added purity data to the supplementary Figure S1. The purity was generally above 90%. We have also added the following sentence to the revised manuscript (line 337-338).

“The purity levels are shown in Supplementary Figure S1.”

Comment 7: Line 352: Specify the method used for lysing MSCs. This detail is critical for reproducibility, as the lysis method could influence downstream experimental outcomes.

Response 7: In response to this comment, we have changed and highlighted the following sentence in the revised manuscript (line 359-360).

“MSCs were lysed on ice for 10 min with cell lysis buffer according to the manufacturer’s instruction (Cell Signaling Technology, Danvers, MA, USA).”

Comment 8: Did you observe any effect of MSC treatment on splenomegaly, given that you performed phenotypic analyses of the spleens? Splenomegaly is a key feature in SLE and addressing this would add value to your study.

Response 8: Yes. In response to this comment, we added the new data to Figure 4F and changed manuscript as follows (line 140-143 & 161).

“The average spleen weight of 22-week-old MRL.Faslpr mice was average 570 mg (indicative splenomegaly), which was slightly reduced by naïve MSCs and significantly reduced by I3A-MSCs (Figure 4F).

Comment 9: Line 222: Replace the word "better" with "more" in this context, as "more" is a more appropriate descriptor for the intended meaning.

Response 9: We have changed it.